# Omega-3 supplements in the prevention and treatment of youth depression and anxiety symptoms: A scoping review

Natalie M. Reily *, Samantha Tang, Ashlee Negrone, Daniel Z. Q. Gan, Veronica Sheanoda, Helen Christensen

Black Dog Institute, University of New South Wales Sydney, Sydney, NSW, Australia

* natalie.reily@blackdog.org.au

## Abstract

### Introduction

Omega-3 supplements may be efficacious in reducing symptoms of depression and anxiety in adults, particularly as an adjunct to antidepressant medication. However, research in young people is limited. Thus, this scoping review aimed to summarise existing evidence on the efficacy of omega-3 supplementation in treating depression and anxiety symptoms in young people aged 14–24. A secondary aim was to determine whether grey literature intended for the general public accurately reflects the evidence.

### Method

Four databases (Cochrane CENTRAL, EmBASE, PsycINFO, PubMed) were searched from inception to 4th August 2021. Eligible peer-reviewed studies were empirical studies which examined the efficacy of omega-3 supplements in preventing/treating anxiety and/or depression symptoms in young people aged 14–24. Risk of bias was assessed for randomised studies using the Cochrane Risk of Bias Tool. Selected grey literature databases were also searched, with eligible sources assessed for quality. A stakeholder group including young people with lived experience of anxiety/depression, parents/carers and mental health professionals informed the research questions and data interpretation. Findings were summarised using narrative synthesis.

### Results

17 empirical studies (N = 1240 participants) meeting inclusion criteria were identified. Studies varied in treatment and participant characteristics. In general, the data did not support the view that omega-3 supplements were efficacious in improving symptoms of anxiety or depression in young people aged 14–24. In contrast, most grey literature sources recommended the use of omega-3 supplements in young people.

**Data Availability Statement:** Given that this is a systematic and scoping review, there is no primary data associated with this work. All included academic articles and grey literature sources are

cited in the manuscript. All data extracted in the review process and during quality assessment is presented within the manuscript or supplementary materials.

**Funding:** This work was funded by a Wellcome Trust Mental Health Priority Area "Active Ingredients" 2021 commission awarded to NMR at Black Dog Institute. HC is supported by a National Health and Medical Research Council Fellowship (1155614). https://wellcome.org/grant-funding https://www.nhmrc.gov.au/ The funders had no role in study design, data collection and analysis, decision to publish, or preparation of the manuscript.

**Competing interests:** The authors declare no competing interests.

## Discussion

Evidence for efficacy of omega-3 supplementation in reducing symptoms of depression and anxiety in young people was inconclusive. More research is needed to identify potential mechanisms and moderators of the effect of omega-3 supplements on depression and anxiety symptoms in young people.

## Introduction

Depression and anxiety are among the most common mental illnesses in young people, with 50% of lifetime cases of mental illness beginning before age 14, and 75% beginning before age 24 [1]. There is also significant comorbidity between depression and anxiety, and other mental health disorders [2]. Standard evidence-based treatments for depression and anxiety in young people typically involve cognitive behavioural therapy, pharmacotherapy, or a combination of both [3, 4]. However, a significant minority of young people do not benefit from such treatments, leading to high rates of relapse [5–7]. This highlights the need to explore other treatments, which can be offered as an alternative or adjunct to standard treatments.

Omega-3 polyunsaturated fatty acids, found in oily fish, flax seeds, walnuts, and oils such as canola and walnut oils, have gained attention for their potential in reducing symptoms of depression and anxiety both in research (for recent meta-analyses see [8–10]) and in the media [11]. Recognition of poor diet quality as a modifiable risk factor implicated in psychological illnesses [12, 13] has led to increasing interest in whether dietary supplements such as omega-3 supplements may be beneficial in preventing or treating common mental health disorders [14]. Over-the-counter omega-3 fish oil capsules typically contain 300mg-600mg of a combination of docosahexaenoic acid (DHA) and eicosapentaenoic acid (EPA) in varying ratios, as well as small amounts of other long-chain omega-3s such as docosapentaenoic acid. There is evidence to show that omega-3 fatty acids interact with a variety of physiological processes implicated in common mood disorders such as the inflammatory response and the regulation of neurotransmitters such as serotonin. Specifically, omega-3 suppresses the upregulation of several proinflammatory cytokines and eicosanoids and related lipid mediators, which may reduce chronic inflammation, a known risk factor for depression [15–17]. Omega-3 supplements may also reduce depression symptoms through its upregulation of serotonin activity [18], given that serotonin pathways are implicated in mood regulation [19].

Several reviews and meta-analyses have examined the efficacy of omega-3 supplements in the prevention and treatment of depression and anxiety in adults. Many have reported that omega-3 supplements are efficacious in treating depression [8, 20–23], and the International Society for Nutritional Psychiatry endorses it as a treatment for depression [24]. However, one recent meta-analysis found that the evidence is imprecise, and concluded that the benefits of omega-3 as a treatment for depression in adults are likely to be non-clinical in magnitude [25]. Additionally, a meta-analysis of four studies examining omega-3 supplements as an adjunct to sertraline found no effect on depressive symptoms [26] and another meta-analysis in adults aged 60 or above had mixed findings [27]. For anxiety, the efficacy of omega-3 supplements in adults has been explored to a lesser extent. However, one meta-analysis found that omega-3 supplements were effective for adults with anxiety symptoms, particularly for those with clinical diagnoses [28]. Regarding prevention of depression and anxiety, a recent meta-analysis suggested that omega-3 supplements had little or no effect on risk of developing depression or anxiety in healthy adult populations [10].

As compared to research in adults, there is substantially less research on whether omega-3 supplements can prevent and treat depression and anxiety symptoms in young people, aged 14–24. A recent meta-analysis of four studies of children aged 6–18 found no evidence of efficacy for omega-3 supplements as a stand-alone depression treatment [9]. However, this review was narrow in scope and excluded non-randomised controlled trials, studies examining anxiety, studies conducted in non-clinical populations, studies that assessed omega-3 as an adjunct rather than a primary, standalone treatment, and studies conducted in clinical samples with mental health disorders other than depression. It is of interest to examine how omega-3 supplements affect depression and anxiety symptoms in the context of other mental health disorders given the prevalence of comorbidity among people with a mental health disorder [2]. An upcoming Cochrane review with a similar scope also stands to investigate the efficacy of omega-3 supplementation for children and adolescents aged 6–19 years (for protocol see [29]).

Critically, there is a gap in knowledge as to the efficacy of omega-3 supplements in young people aged 14–24 –no systematic reviews or meta-analyses, to our knowledge, have focused specifically on this age group, despite their high risk of depression and anxiety. While there are some guidelines on the use of omega-3 in adult depression [24], there are no such guidelines for young people. The primary aim of this review was therefore to synthesise the current literature on the efficacy of omega-3 supplements for depression and anxiety symptoms in young people, including potential mechanisms of action and moderators of efficacy. A secondary aim and novel contribution to the literature was to investigate whether grey literature commentary aimed at the general public on omega-3 supplements for depression and anxiety symptoms in youth was consistent with the available evidence from scientific literature. Given the two aims, a scoping review approach was taken which also incorporated lived experience input in its design and interpretation of findings. Lived experience can aid in facilitating translatable and human-centered research in the mental health sphere [30].

## Method

### Protocol

We followed the Preferred Reporting Items for Systematic Reviews and Meta-Analyses extension for Scoping Reviews (PRISMA-ScR) Checklist [31]; see S1 Table for the PRISMA-ScR Checklist. The protocol was registered with the Open Science Framework (OSF.IO/WFB7D).

### Lived experience stakeholder consultation

A group of 11 stakeholders—consisting of four young people with lived experience of depression and/or anxiety, three parents of young people with lived experience of depression and/or anxiety, and four healthcare professionals—were recruited online through social media (e.g., Facebook) to share their insights and perspectives during two online workshops. After seeing the online advertisement, potential stakeholders filled out an expression of interest form which included questions about demographic characteristics (e.g., age, gender), profession (for health professionals only) and asked about their availability to attend various potential workshop dates. Stakeholders were selected on the basis of their availability, and to ensure there was as much diversity as possible amongst the group in terms of age, gender, profession, and experience with using omega-3 or other dietary supplements. Following our organisations' Lived Experience Policy, ethics approval was not sought given that lived experience stakeholders were engaged in an advisory capacity. However, all stakeholders provided written informed consent to attend advisory workshops. These workshops were held on Zoom (San Jose, California) and used the online collaborative tool Miro (San Jose, California). The first workshop, held on 13th July 2021, prior to database searching, sought stakeholders' input on (i) types of

information sources from which they would seek information about the efficacy of omega-3, (ii) what they were interested in understanding about omega-3 supplementation for depression and anxiety symptoms, and (iii) relevant search terms that could be used in the search strategy. The second workshop was held on 13th September 2021, after data were extracted from randomised controlled trials (RCTs) identified in the search. The purpose of this workshop was to obtain input on data interpretation, insights on how to use grey literature and ways of communicating study findings.

### Search strategy and selection criteria

During the first workshop, stakeholders expressed interest in understanding the effectiveness of omega-3 supplements in preventing and treating depression and anxiety symptoms. Thus, we systematically searched four online academic databases for articles published from database inception to 4th August 2021: Cochrane CENTRAL EmBASE (from 1947), PsycINFO (from 1806), and PubMed (from 1996). The search comprised of three blocks of search terms organised around (i) omega-3 polyunsaturated fatty acids, (ii) young people, and (iii) depression and anxiety (see S2 Table for search terms used in each database).

We adapted the above strategy to search the grey literature (see S3 Table). Government and health authority databases were selected from the Canadian Agency for Drugs and Technologies in Health (CADTH) grey literature checklist [32]. A Google advanced search was also conducted and results of the first 10 pages were extracted.

Eligible peer-reviewed studies met the following inclusion criteria: (i) mean participant age between 14 and 24 years, (ii) administered omega-3 supplements containing DHA and EPA, (iii) measured anxiety and/or depression symptoms using validated measures, (iv) published in the English language, and (v) contained empirical data (i.e., reviews, commentaries and case studies were excluded). Both non-clinical and clinical samples were included, with no restrictions placed on psychiatric diagnoses. Studies that administered omega-3 supplements as an adjunct to treatment as usual were included, with no restrictions placed on the type of usual treatment. Correlational studies examining intake of fish and mental health outcomes or biomarkers of omega-3 in the body and mental health outcomes were excluded. Studies that altered components of diet (e.g., amount of fish consumed) rather than delivering tablets/supplements were also excluded as it was not possible to precisely measure omega-3 dosage.

Grey literature information sources were eligible if they met the following criteria: (i) described the impact of omega-3 on depression and/or anxiety in young people, and (ii) targeted potential consumers, policymakers, or health professionals. Sources that targeted academic audiences (e.g., clinical trial protocols) were excluded.

### Screening

Two authors (DZQG and NMR or AN) independently screened the titles and abstracts identified through peer reviewed literature searches, and the titles only for the grey literature searches for a subset of 10% of the articles, with disagreements resolved through discussion. All remaining titles and abstracts were screened by DZQG. At the full-text stage, all articles were independently screened by two authors (DZQG, NMR, ST, AN; Cohen's kappa = 0.58–0.87), with disagreements resolved through discussion, and a third author consulted if consensus could not be reached.

### Data extraction

Data extracted from the peer-reviewed literature included: authors and year of publication, country, sample characteristics (size, age, gender, diagnosis), study type, dosage and duration

of omega-3 supplementation, other treatments administered, outcomes assessed, main findings, and information on side effects and compliance. Data extracted from the grey literature included: authors, publisher, and year of publication, country, target audience, and key messages on the use of omega-3 for depression and anxiety in young people. Two authors (AN and DZQG) independently extracted and coded data sources using Covidence (Veritas Health Innovation, 2020) for all eligible peer reviewed articles and a customised Microsoft Excel spreadsheet for grey literature sources. Disagreements were resolved through discussion. Corresponding authors of studies were contacted by email if more information was needed to determine eligibility.

## Quality assessment

Risk of bias of included studies is not typically assessed in scoping reviews [31]. Nonetheless, we conducted quality assessment ratings of the included RCTs using Version 2 of the Cochrane risk-of-bias tool for randomised trials [33]. This tool assesses possible sources of bias in RCTs, including: (1) randomisation sequence generation and allocation concealment, (2) blinding of participants, personnel and outcome assessors, (3) incomplete outcome data, (4) measurement of outcome and (5) selective reporting. Risk of bias ratings for RCTs (see S4 Table) were independently performed by AN and DZQG, and disagreements were resolved through discussion.

Grey literature information sources were also assessed by two raters (AN and DZQG) on three criteria: comprehensiveness, accuracy of information, and the extent to which references to the peer-reviewed literature were incorporated. The criteria selected were informed by the stakeholder group and adapted from a previous rating system developed by Wade et al. [34]. Each criterion was rated *Poor*, *Moderate*, or *Excellent* (see S5 Table).

## Synthesis of results

Significant variability in the design of included studies precluded a meta-analytic approach. A narrative synthesis approach was therefore undertaken to summarise findings in relation to each identified variable.

## Results

### Peer-reviewed literature

**Study characteristics.**   The search yielded a total of 5264 articles. Following removal of duplicates and screening, 17 studies met inclusion criteria (see Fig 1), of which 13 were RCTs. Table 1 displays key characteristics of each included RCT study. All examined depression symptoms as an outcome, but only five measured anxiety symptoms. Studies varied in the daily dosage of omega-3 administered (1000mg/day-6400mg/day of EPA and DHA combined), the ratio of EPA to DHA in each dosage (50% EPA-85.7% EPA), and duration of treatment (3 weeks-12 months). Most studies were conducted in the United States (n = 7), and only one study was conducted in a low- to middle-income country (Iran).

Of the 13 RCTs, six administered omega-3 supplements as an adjunct to pharmacological or psychosocial interventions. Two RCTs used a combination of omega-3 and vitamin supplements. The remaining five RCTs administered omega-3 supplements exclusively. Sample populations varied across studies with the majority in clinical samples including people with depression (n = 4), and people with, or at risk of, a mental illness other than depression or anxiety, including psychosis, schizophrenia, bipolar disorder, borderline personality disorder, and anorexia (n = 6). The remaining three studies used non-clinical samples, described as healthy

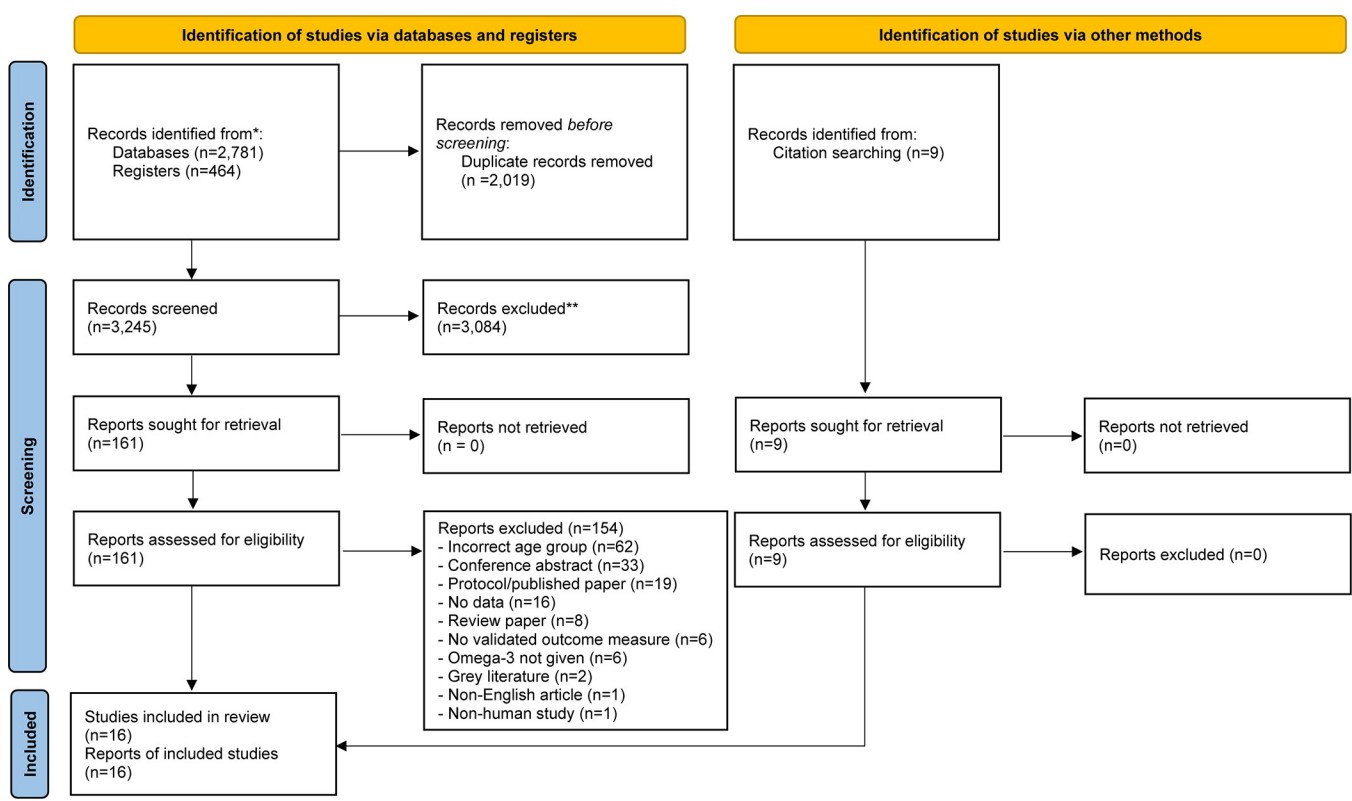

**Fig 1. PRISMA flow diagram: Scientific literature screening process.**

individuals, women with polycystic ovarian syndrome, and medical students respectively. No RCTs were focused specifically on identifying moderators of efficacy or differentiating between potential mechanisms of action. Table 2 presents a summary of results of all included RCTs.

In addition to the thirteen RCTs, there were two open label trials [35, 36] and one observational study [37]. Finally, one other report identified in the grey literature search was included with these additional studies because it reported on empirical data, despite not being peer-reviewed [38]; see S6 Table for non-RCT characteristics.

**Efficacy of omega-3 supplements on depression.** Two RCTs in young people with depression found that omega-3 supplementation led to greater reductions in depressive symptoms relative to placebo. These studies differed in dosage and duration of treatment, with one administering 1400mg/day for 3 weeks as a standalone treatment [39] and the other administering 2400mg/day for 12 weeks in addition to standard antidepressant therapy [40]. One of these studies also found that depressed participants had significantly lower baseline levels of EPA and DHA compared to healthy controls [40]. A third study in young women with polycystic ovarian syndrome found that a low dose of 1000mg/day of omega-3 co-supplemented with vitamin E led to lower levels of depressive symptoms relative to the placebo control group [41]. The remaining ten RCTs found null effects [42–51]. In addition to finding no significant effect on general symptoms of depression, one study also found no effect on specific symptoms of depression, including irritability, suicidality and anhedonia [44].

The effect of omega-3 supplementation was also examined by four non-randomised controlled trials in young people with depression (n = 3) and bipolar disorder (n = 1). All four studies found omega-3 supplementation reduced depressive symptoms over time, however, none included an adequate placebo control (see S7 Table).

**Table 1. Characteristics of randomised controlled trials investigating the effect of omega-3 supplementation on depression and/or anxiety (n = 13).**

| Study ID | Country | Sample size, n | Female participants (%) | Demographic | Diagnostic criteria | Age, mean (SD) | Control | Outcome measure |
|---|---|---|---|---|---|---|---|---|
| **Amminger, 2010** [42] | Austria | 81 | 66.7 | Adolescents with subthreshold psychotic disorder | DSM-IV | Omega-3 = 16.8 (2.4) Placebo = 16 (1.7) | Placebo | MADRS, GAF, PANSS, fasting erythrocyte fatty acid composition |
| **Amminger, 2013** [43] | Austria | 15 | 93.3 | Adolescents with BPD and UHR for psychosis | DSM-IV | 16.2 (2.1) | Placebo | MADRS, GAF, erythrocyte fatty acid composition |
| **Gabbay, 2018** [44] | USA | 51 | 48.3 | Adolescents with depression of ≥6 weeks, minimum raw score of 40 for CDRS-R | DSM-IV | 16.1 (2.1) | Placebo | CDRS, BDI-II, BSS |
| **Giles, 2015** [45] | USA | 72 | 62.5 | Healthy individuals | None | Omega-3 = 20.8 (2.4) Placebo = 20.5 (1.7) | Placebo | POMS, STICSA, TSST EIT, MFT, IL-1β, salivary cortisol |
| **Ginty, 2015** [39] | USA | 23 | 78.0 | Young adults with depression | BDI score >10 | 20.2 (1.3) | Placebo | BDI |
| **Jamilian, 2018** [41] | Iran | 40 | 100.0 | Women with PCOS | None | Omega-3 = 22.3 (4.7) Placebo = 24.4 (4.7) | Placebo | BDI, GHQ-28, DASS42, PPAR-γ, IL-1, GLUT-1, TNF-α, TGF-β, hirsutism |
| **Kiecolt-Glaser, 2011** [46] | USA | 68 | 44.1 | Medical students | None | 23.65 (1.9) | Placebo | CESDS, BAI, PSQI, SDPAR, side effects, fatty acid analyses, serum and stimulated IL-6 and TNF-α, SAD |
| **Manos, 2018** [47] | USA | 24 | 100.0 | Adolescents with anorexia nervosa | DSM-V | Omega-3 = 15.0 (1.3) Placebo = 14.4 (1.8) | Placebo | BAI, CESDS, EAT, BMI, medication side effects |
| **McGorry, 2017** [48] | Australia, Asia, Europe | 304 | 54.3 | Young people at UHR for psychotic disorders | DSM-IV | 19.1 (4.6) | Placebo | SANS, MADRS, GFSRS, CAARMS, BPRS, YMRS, SOFS |
| **McNamara, 2020** [49] | USA | 56 | 80.0 | Adolescents at high risk of bipolar disorder | DSM-IV, CDRS-S score ≥40 | 14.1 (3.0) | Placebo | CDRS-R, CGI-S, CGI-I, CGAS, YMRS, ADHD-RS, CBCL, fatty acid composition, proton magnetic resonance spectroscopy, side effects |
| **Robinson, 2019** [51] | USA | 50 | 30.0 | Patients diagnosed with recent onset psychosis, schizophrenia or bipolar disorder treated with risperidone | DSM-IV | Omega-3 = 21.1 (5.3) Placebo = 22.0 (5.3) | Placebo | SANS, BPRS, CGI-S, metabolic outcomes*, motor outcomes |
| **Trebatická, 2020** [40] | Slovakia | 60 | 80.0 | Children with depression | ICD-10 | 15.7 (1.6) | Omega-6 active control | CDI, serum fatty acid levels, ratio of omega-3:omega-6 |
| **Van der Wurff, 2020** [50] | Netherlands | 267 | 52.0 | Adolescents with depression | CESDS score | Omega-3 = 14.2 (0.5) Placebo = 14.1 (0.5) | Placebo | CESDS, RSEQ, blood fatty acids, omega-3-index |

ADHD-RS = Attention-Deficit with Hyperactivity Disorder Rating Scale; BAI = Beck Anxiety Inventory; BDI = Beck Depression Inventory; BPRS = Brief Psychiatric Rating Scale–Anchored version; BSS = Beck Scale for Suicide Ideation; BPD = Borderline Personality Disorder; CAARMS = Comprehensive Assessment of the At-Risk Mental State; CBCL = Child Behaviour Checklist; CDI = Children's Depression Inventory; CDRS-R = Children's Depression Rating Scale Revised; CESDS = Centre for Epidemiological Studies Depression Scale; CGAS = The Children's Global Assessment Scale; CGI-I = Clinical Global Impression- Improvement Scale; CGI-S = Clinical Global Impression-Severity Scale; DASS = Depression Anxiety and Stress Scale; DHA = docosahexaenoic acid; DSM-IV = Diagnostic and Statistical Manual of Mental Disorders–Fourth Edition; EAT = Eating Attitudes Test; EIT = Emotional Interference Task; EPA = eicosapentaenoic acid; FHRDC = Family History Research Diagnostic Criteria; GAF = Global Assessment of Functioning; GFSRS = Global Functioning: Social and Role Scale; GLUT-1 = glucose transporter-1; GHQ-28 = General Health Questionnaire 28; ICD-10 = International Classification of Diseases; IL-1 = interleukin-1; IL-6 = interleukin 6; IL-8 = interleukin-8; MADRS = Montgomery-Asberg Depression Rating Scale; MADD = Mixed Anxiety and Depressive Disorder; MDD = Major Depressive Disorder; MFT = Morphed Faces Task; OATS = Omega-3 and Therapy Study; PANSS = Positive and Negative Syndrome Scale; PCOS = polycystic ovarian syndrome; POMS = Profile of Mood States Questionnaire; PSQI = Pittsburg Sleep Quality Index; RSEQ = Rosenberg Self Esteem Questionnaire; QIDS = Quick Inventory of Depressive Symptomatology; SAD = sagittal abdominal fat; SANS = Schedule for Assessment of Negative Symptoms, Hillside Clinical Trials version; SDPAR = Seven Day Physical Activity Recall; SOFS = Social and Occupational Functioning Scale; STICSA = State-Trait Inventory for Cognitive and Somatic Anxiety; TNF-α = tumour necrosis factor alpha; TGF-β = transformative growth factor beta; TSST = Trier Social Stress Test; UHR = Ultra High Risk; YMRS = Young Mania Rating Scale

* = metabolic outcomes included weight, body mass index, total cholesterol, low-density lipoprotein cholesterol, high-density lipoprotein cholesterol, triglycerides, haemoglobin, fasting glucose

Table 2. Results from randomised controlled trials investigating the effect of omega-3 supplementation on depression and/or anxiety (n = 13).

| Study ID | Omega3^ daily dose | % EPA, DHA | Other interventions | Depression* | Anxiety* | Other outcomes* | Side effects and adherence |
|---|---|---|---|---|---|---|---|
| **Amminger 2010 [42]** | 1800mg + 7.6mg vitamin E for 12 weeks | 59.3% EPA 40.7% DHA | Antidepressants: omega-3 (12.2%); placebo (7.5%) Benzodiazepines: omega-3 (4.9%); placebo 2.5% of placebo | No significant change in depression (MADRS) | Not studied | Significantly increased global functioning (GAF), and reduced positive, negative and general psychosis symptoms (PANNS) and risk of transition to psychotic disorder. Significant increase in omega-3 relative to omega-6 in treatment group. Change in this ratio in treatment group was significantly associated with functional improvement (GAF). | No significant differences in side effects between groups. Mean adherence: omega-3 (81.4%); placebo (75.4%) |
| **Amminger 2013 [43]** | 1180mg + 7.6mg vitamin E for 12 weeks | 59.3% EPA 40.7% DHA | Antidepressants: omega-3 (25.0%); placebo (14.3%) Benzodiazepines: omega-3 (12.5%); placebo (14.3%) Counselling: M = 7 sessions (both groups) | No significant change in depression (MADRS) | Not studied | Significantly improved PANSS negative, general, BPD and total scores compared to placebo. No significant difference on PANSS positive subscale or in global functioning (GAF). | No clinically relevant side effects observed, or group differences in side effects or adherence: omega-3 (77.1%); placebo (87.2%) |
| **Gabbay 2018 [44]** | 1200mg-3600mg in increments of 600mg per day, for 10 weeks | 66.7% EPA 33.3% DHA | None | No significant difference in depression (CDRS, BDI-II), with improvements for both treatments | Not studied | No significant change in irritability, suicidality or anhedonia (BSS, BDI-II, CDRS) | No significant difference in side effects or adherence between groups |
| **Giles 2015 [45]** | 2800mg for 5 weeks | 60% EPA 40% DHA | None | No significant change in depression (POMS) | No significant change in somatic or cognitive anxiety (STICSA) | No significant effect in EIT, MFT levels/scores in omega-3 as a main effect or in interaction with stress (TSST). No significant change in IL-1β, cortisol, as an interaction with stress in main effect. | No clinically relevant side effects observed. Mean adherence: omega-3 (99.6%); placebo (99.3%) |
| **Ginty 2015 [39]** | 1400mg for 3 weeks | 50% EPA 50% DHA | None | ↓ depression symptoms (BDI) to below clinically indicated levels: omega-3 (67.0%); placebo (20.0%), significant interaction over time. ↓ cognitive-affective symptomology over time. No difference in somatic-vegetative symptomology (BDI) | Not studied | Not studied | Not reported |
| **Jamilian 2018 [41]** | 1000mg omega-3 for 12 weeks | Not reported | 400IU Vitamin E daily | ↓ significant improvements in depressive symptoms (BDI) | ↓ depression/ anxiety (DASS42) | ↓ general health symptoms (GHQ-28), hirsutism. No significant difference between groups in TGF-β or salivary cortisol. ↓ serum insulin levels and HOMA-IR, IL-8 and TNF-α expression. ↑ PPAR- γ. Did not change GLUT-1 or TGF-β expression in PBMC. | Side effects not reported, adherence of >90% capsules taken in both groups |

(Continued)

**Table 2.** (Continued)

| Study ID | Omega3^ daily dose | % EPA, DHA | Other interventions | Depression* | Anxiety* | Other outcomes* | Side effects and adherence |
|---|---|---|---|---|---|---|---|
| Kiecolt-Glaser 2011 [46] | 2496mg for 12 weeks | 85.7% EPA 14.3% DHA | Multivitamins permitted | No significant difference in depression (CESDS, average of 4 time points) | ↓ 20% reduction in anxiety symptoms compared to placebo (BAI). | No significant difference in self-reported dietary intake or physical activity (SDPAR). Decreased geometric mean scores of LPS-stimulated IL-6 in Omega-3 group. No effect on serum IL-6 or TNF-α. Significant negative correlation between plasma omega-3 levels and anxiety (BAI). Decreasing omega-6 to omega-3 ratios led to lower anxiety, stimulated IL-6 and TNF-α (estimated). Omega-3 to omega-6 ratios did not affect depressive symptoms or serum IL-6, and had a borderline significant effect on TNF-α. No effect of gender or SAD. | No significant difference in side effects or adherence (95.6% overall) between groups |
| Manos 2018 [47] | 3124mg for 12 weeks | 67.9% EPA 19.2% DHA 12.9% other PUFAs | 87.5% of participants on SSRIs | No significant change in depression (CES-D) | No significant change in anxiety (BAIT) | No significant difference in EAT score, BMI, or medication side effects. | No group differences in side effects, adherence not reported |
| McGorry 2017 [48] | 1400mg for 6 months | 60% EPA 40% DHA | 100% cognitive-behavioural case management. SSRIs, benzodiazepines permitted | No significant change in depression | Not studied | No significant difference in transition to psychosis or change in psychotic symptoms (BPRS; SANS; YMRS), and global functioning (SOFAS; GFSRS) | No group differences in side effects or mean adherence: omega-3 (43.1%); placebo (41.4%) |
| McNamara 2020 [49] | 2130mg for 12 weeks | 60% EPA 5.3% DPA 34.6% DHA** | None | No significant group differences in depression (CDRS-R). Similar remission rates in both placebo and omega-3. | Not studied | ↑ global functioning scores (CGAS). ↓ CGI-S and CGI-1 scores. No group differences in mania (YMRS), ADHD (ADHD-RS) or CBCL. No significant difference to pre-post treatment in lipids, liver function, TSH, WBCs, RBCs or platelets. ACC creatinine and ACC choline differed significantly between groups from baseline to endpoint. Baseline ACC choline levels were inversely correlated with baseline to endpoint changes in CDRS-R scores. ACC Cho baseline to endpoint change correlated with CDRS-R scores and Omega-3 levels. No significant metabolite difference in right and left VLPFC. | More muscle cramps experienced by omega-3 group. No group differences other side effects. Adherence not reported. |
| Robinson 2019 [51] | 1140mg omega-3 + 2mg/g tocopherol daily for 16 weeks | 64.9% EPA 35.1% DHA | 100% on risperidone, 54% on lorazepam. Benztropine mesylate, propranolol, zolpidem or rozerem permitted | In participants not taking lorazepam, significant tmt-by-time interaction favouring omega-3 on the depression-anxiety factor. No significant change in depressed mood for participants not taking lorazepam (BPRS) | ↓ anxiety symptoms for participants not taking lorazepam (BPRS) | ↓ BPRS scores for participants not taking lorazepam. No significant change in motor outcomes or metabolic outcomes. No significant difference in SANS global scores. | Adverse events greater for placebo than omega-3 group, but no analyses conducted. Mean adherence of 50% in both groups. |

(*Continued*)

**Table 2.** (Continued)

| Study ID | Omega3^ daily dose | % EPA, DHA | Other interventions | Depression* | Anxiety* | Other outcomes* | Side effects and adherence |
|---|---|---|---|---|---|---|---|
| **Trebatická 2020 [40]** | 2400mg for 12 weeks | 57.1% EPA 42.9% DHA | Standard antidepressant therapy | ↓ depression scores (CDI) in omega-3 group, with greater improvements in DD than MADD. | Not studied | Significantly lower levels of EPA, DHA, AA + LA in depressed patients. Unchanged levels of C16+C18 in depressed patients. Ratio of omega-6:omega-3 decreased in treatment group after 6 and 12 weeks. EPA and omega-6: omega-3 ratio correlated with severity symptoms at baseline. No significant difference between males and females or between MADD and DD. | No serious adverse events. No significant difference in side effects between groups, >95% adherence in both groups. |
| **Van der Wurff 2020 [50]** | Low dose: 1600mg for 12 months, up to 3200mg after 3 months High dose: 6400mg for 12 months. | 65% EPA 35% DHA | None | No significant change in depression (CESDS) | Not studied | No significant change in self-esteem (RSEQ). Significantly higher concentrations of EPA, DPA, DHA; significantly lower concentrations of AA and ObA in treatment group. Higher omega-3 index in treatment group, however no association found with depression score. | Side effects not reported. Adherence of 56.1% across both groups. |

AA = arachidonic acid; ACC = Anterior cingulate cortex; ADHD-RS = Attention-Deficit with Hyperactivity Disorder Rating Scale; BAI = Beck Anxiety Inventory; BDI (II) = Beck Depression Inventory; BPRS = Brief Psychiatric Rating Scale–Anchored version; BSS = Beck Scale for Suicide Ideation; BPD = Borderline Personality Disorder; C16 = palmitic acid; C18 = stearic acid; CAARMS = Comprehensive Assessment of the At-Risk Mental State; CBCL = Child Behaviour Checklist; CDI = Children's Depression Inventory; CDRS-R = Children's Depression Rating Scale Revised; CESDS = Centre for Epidemiological Studies Depression Scale; CGAS = The Children's Global Assessment Scale; CGI-I = Clinical Global Impression- Improvement Scale; CGI-S = Clinical Global Impression-Severity Scale; DASS = Depression Anxiety and Stress Scale; DHA = docosahexaenoic acid; DSM = Diagnostic and Statistical Manual of Mental Disorders; EAT = Eating Attitudes Test; EIT = Emotional Interference Task, EPA = eicosapentaenoic acid; FHRDC = Family History Research Diagnostic Criteria; GAF = Global Assessment of Functioning; GFSRS = Global Functioning: Social and Role Scale; GLUT-1 = glucose transporter-1; GHQ-28 = General Health Questionnaire 28; HOMA-IR = homeostatic model assessment for insulin resistance; ICD-10 = International Classification of Diseases; IL-1 = interleukin-1; IL-6 = interleukin 6; IL-8 = interleukin-8; MADRS = Montgomery-Asberg Depression Rating Scale; MADD = Mixed Anxiety and Depressive Disorder; MDD = Major Depressive Disorder; MFT = Morphed Faces Task; OATS = Omega-3 and Therapy Study; ObA = osbond acid; PANSS = Positive and Negative Syndrome Scale; PCOS = polycystic ovarian syndrome; POMS = Profile of Mood States Questionnaire; PSQI = Pittsburg Sleep Quality Index; PUFAs = polyunsaturated fatty acids; RBC = red blood cells; RSEQ = Rosenberg Self Esteem Questionnaire; QIDS = Quick Inventory of Depressive Symptomatology; SAD = sagittal abdominal fat; SANS = Schedule for Assessment of Negative Symptoms, Hillside Clinical Trials version; SDPAR = Seven Day Physical Activity Recall; SOFS = Social and Occupational Functioning Scale; STICSA = State-Trait Inventory for Cognitive and Somatic Anxiety; TNF-α = tumour necrosis factor alpha; TGF-β = transformative growth factor beta; TSH = thyroid stimulating hormone; TSST = Trier Social Stress Test; UHR = Ultra High Risk; VLPFC = ventrolateral prefrontal cortex; YMRS = Young Mania Rating Scale

↑ denotes improvement; ↓ denotes decrease in symptoms

* denotes result following intervention and statistically significant according to test parameters

** percentage does not total to 100% due to rounding

^Dosage in mg refers to total long chain omega-3 fatty acids (i.e., EPA and DHA only).

Overall, there is weak evidence to suggest that omega-3 supplements are effective in reducing depressive symptoms among young people diagnosed with depression or other mental illnesses, or in non-clinical populations.

**Efficacy of omega-3 supplements on anxiety.** Of the five RCTs that assessed anxiety outcomes, three found that omega-3 reduced anxiety symptoms, including two studies in non-clinical populations [41, 46] and one study in young people taking risperidone for recent onset psychosis, schizophrenia, or bipolar disorder [51]. Dosage and duration of treatment was varied, with one administering 1000 mg/day with vitamin E for 12 weeks [41], another administering 2496 mg/day for 12 weeks as a standalone treatment [46], and the third administering 1140mg omega-3 with tocopherol in addition to patients' standard treatments for recent onset psychosis [51]. One of these studies also found plasma omega-6 to omega-3 ratios were positively correlated with anxiety symptoms after supplementation [46]. Two studies found no effect of omega-3 supplementation on anxiety symptoms, including one in a non-clinical sample [45] and one in adolescents with anorexia nervosa [47].

**Secondary outcomes (inflammation, metabolism, other mental health symptoms).** Two studies that found an effect of omega-3 supplementation on anxiety symptoms [46], or both anxiety and depression symptoms [41] also found corresponding changes in inflammatory biomarkers. Specifically, Kiecolt-Glaser et al. [46] found that omega-3 supplementation reduced stimulated interleukin-6 levels (IL-6), but not serum IL-6 levels or tumor necrosis factor alpha (TNF-α) production, and Jamilian and colleagues [41] found that omega-3 supplementation downregulated interleukin-8, TNF-α and serum insulin, and upregulated peroxisome proliferator-activated receptor expression. A third study, which reported a reduction in anxiety symptoms with omega-3 supplementation, found no between-group differences in metabolic outcomes including weight, BMI, cholesterol, triglycerides, haemoglobin levels, and fasting glucose [51].

Other outcomes assessed included psychosis-related symptoms [42, 43, 48, 51], general psychological distress [41], global functioning [42, 48], self-esteem [50], and neural activity [49]. These secondary outcomes are reported in Table 2.

**Side effects and adherence to intervention.** There was little evidence of side effects attributable to omega-3 supplementation. One study reported that participants receiving omega-3 supplements were more likely to experience muscle cramps compared to placebo [49], whereas another observed a greater number of adverse events in the placebo group, but no statistical analyses were performed [51]. The remaining studies either found no differences in side effects between omega-3 and placebo groups (n = 8) or did not report on side effects (n = 3). Adherence to the intervention also did not differ significantly between omega-3 and placebo conditions.

**Quality assessment.** Only two of the RCTs had low risk of bias across all 5 domains (see S3 Table). Selective reporting of results was identified to be the largest source of possible bias, with 9 studies (69.2%) not reporting sufficient information (such as a prospectively published trial protocol) to rule out bias in this domain. The second most common source of potential bias was associated with the randomisation process, with 6 studies (46.2%) not reporting sufficient detail on randomisation sequence generation or allocation concealment. Risk of bias relating to the other domains was mostly low.

## Grey literature

**Source characteristics.** Twelve grey literature sources met inclusion criteria, including seven online articles, three blogs, one fact sheet and one practice guideline, all published from 2005–2021 (see Fig 2). The majority were from the United States (n = 7). See Table 3 for grey literature source characteristics.

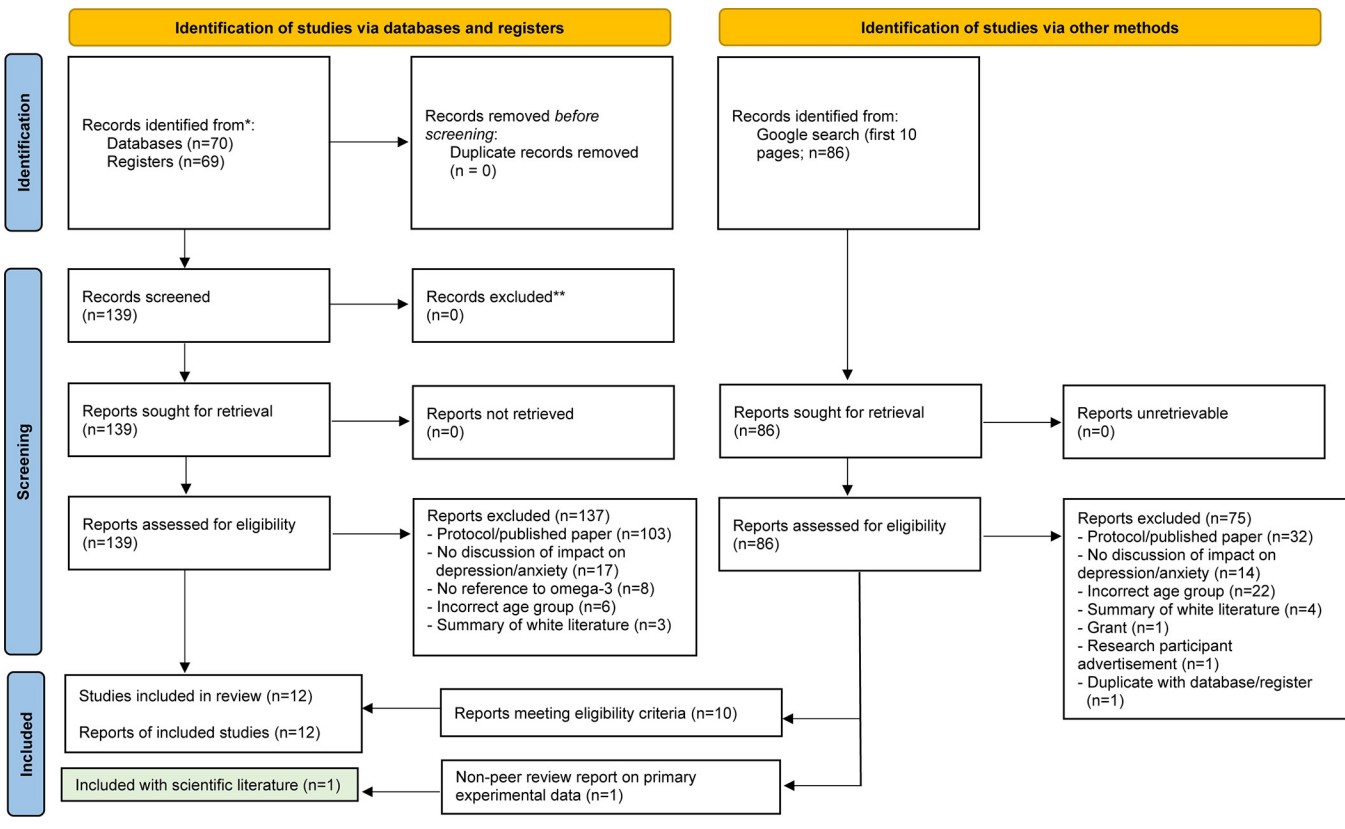

**Fig 2. PRISMA flow diagram: Grey literature screening process.**

**Key messages.** As with the peer-reviewed literature, most grey literature discussed the effect of omega-3 on depression as opposed to anxiety. However contrary to peer-reviewed literature, most sources recommended omega-3 to alleviate depression or anxiety symptoms (83%). Specifically, seven websites recommended omega-3 for low mood or depression [52–58], one for anxiety [59], and two for both anxiety and depression ([60, 61]. The remaining two sources were neutral [62, 63], with none advising against the use of omega-3.

Sources that recommended a specific dosage of omega-3 supplements recommended between 500-2000mg/day with a high ratio of EPA to DHA [52, 56, 60]. However, other sources advised for omega-3 to be obtained through diet rather than in supplement form [55, 59, 61]. Most sources mentioned benefits of omega-3 intake for other aspects of health, such as cognitive ability [53, 61]. Thus, promotion of omega-3 in grey literature often centered around omega-3 as a part of a healthy diet, which was stated to indirectly contribute to better mental health. Side effects of supplementation reported in grey literature were predominately similar to those noted in peer-reviewed literature, including fishy aftertaste, gastrointestinal disturbances [53] and increased risk of bleeding [54, 58].

**Quality assessment.** Quality assessment of grey literature sources revealed variation in accuracy, comprehensiveness, and extent to which peer-reviewed literature was referenced (see S4 Table). Reference to peer reviewed literature was the highest scoring domain, with 58% of grey literature sources rated as excellent. However, despite many sources referencing peer reviewed literature, only 17% of sources were rated as excellent in accuracy. The majority of remaining sources unreservedly promoted the use of omega-3 supplements for as a treatment for depression and anxiety in young people, failing to acknowledge the lack of consistent

**Table 3. Summary of grey literature reporting on the effect of omega-3 supplementation on depression and/or anxiety (n = 12).**

| Author, publisher, year | URL | Type | Country | Target audience | Stance on omega-3 | Key message | Side effects |
|---|---|---|---|---|---|---|---|
| **Amen Clinics, 2020** [60] | https://www.amenclinics.com/blog/new-evidence-on-omega-3s-for-behavioral-problems-in-children/ | Online article | United States | General public | Recommends | Research on the effect of omega-3 on depression in children is "sparse but encouraging". Evidence says supplementation with omega-3 improves quality of life and mental status in children with emotional and behavioural issues. Low levels of EPA and DHA linked to mind disorders such as depression. Recommended dose is 800mg for every 40lb body weight, and ratio of 60/40 EPA:DHA. More research on omega-3 in adults than in children. | Not reported |
| **Bartholomew, Nutri Advanced, 2021** [52] | https://www.nutriadvanced.co.uk/news/top-10-reasons-to-give-your-kids-omega-3/ | Blog | United Kingdom | General public | Recommends | Omega-3 is effective in mitigating symptoms of depression in children and youth. Two studies were referenced that found that omega-3 can improve depression in children by improving blood flow to brain. One study mentioned dose as 60mg per day of EPA and 1560mg DHA. | Not reported |
| **British Diet Association, 2020** [59] | https://www.bda.uk.com/resourceDetail/printPdf/?resource=diet-behaviour-and-learning-children | Fact sheet | United Kingdom | General public | Recommends | There is some evidence omega-3 can improve mood and reduced anxiety in people with ADHD and autism, mostly based on adult studies. Consumption of food rich in omega-3 is recommended for children. | Not reported |
| **Headspace, 2019** [61] | https://headspace.org.au/blog/the-best-foods-for-mental-health/ | Blog | Australia | General public | Recommends | DHA can help boost mood, improve feelings of depression, and reduce anxiety. Omega-3 in fish can help circulate serotonin and dopamine. | Not reported |
| **Heyes, UNICEF, 2015** [62] | https://www.unicef-irc.org/article/1149-the-adolescent-brain-vulnerability-and-opportunity.html | Online article | Italy | Policymakers, government | Neutral | Studies are being conducted currently to evaluate whether omega-3 supplementation during adolescence may alleviate symptoms of emotional disorders. More generally, good physical health has been linked to positive mental health in adolescence (and the reverse). | Not reported |
| **Kemper, Contemporary Pediatrics, 2005** [54] | https://www.contemporarypediatrics.com/view/does-fish-oil-help-adhd | Blog | United States | Healthcare professionals | Recommends | EPA and DHA appear to be helpful for mental health problems such as depression, which is often comorbid with ADHD. Trials are needed for adolescents with ADHD. | Not reported |
| **Link, Healthline, 2019** [53] | https://www.healthline.com/nutrition/omega-3-for-kids | Online article | United States | General public | Recommends | Studies show that omega-3 fatty acids may improve brain function and prevent depression and mood disorders in children. Recommended dosage ranged from 0.5g to 1.6g, varying by age and sex. | Bad breath, unpleasant aftertaste, gastrointestinal disturbances* |
| **Miller, Hey Sigmund, 2017** [55] | https://www.heysigmund.com/resilience-anxiety-and-nutrition/ | Online article | Australia | General public | Recommends | More research has examined the effect of omega-3 on mood and brain health than any other nutrient. Oily fish will help calm anxiety and many other mental health problems by helping to feed the brain. | Not reported |

(Continued)

Table 3. (Continued)

| Author, publisher, year | URL | Type | Country | Target audience | Stance on omega-3 | Key message | Side effects |
|---|---|---|---|---|---|---|---|
| **Newport Academy, 2017** [57] | https://www.newportacademy.com/resources/empowering-teens/food-mood-nutrition-teen-mental-health/ | Online article | United States | General public | Recommends | Omega-3 may improve quality of life and mental status in children with emotional and behavioural issues. Links depression and deficiencies in particular nutrients. Omega-3 plays a role in affecting serotonin and dopamine. Significant improvement in teen depression and nutrition if omega-3 and amino acids are incorporated. | Not reported |
| **Pediatric partners, unknown date** [58] | https://pediatricpartnerskc.com/Education/Nutrition/OMEGA-3-FATTY-ACIDS-FISH-AND-NUT-OILS | Online article | United States | General public | Recommends | Fish oil has benefits for many aspects of health (high blood pressure, high triglycerides, infant brain and eye development, possibly preventing cancers), and can be used to treat mood and behavioural disorders (ADHD and depression). No standard dosing exists, but many healthcare professionals recommend 1000mg EPA+DHA for children and 2000mg for an adult (no source). | Gastrointestinal disturbances*, increased bleeding, vitamin E deficiency, fishy aftertaste |
| **Sparks, Kaiser Permanente, 2021** [56] | https://wa.kaiserpermanente.org/static/pdf/public/guidelines/depression.pdf | Practice guidelines | United States | Healthcare professionals | Recommends | Omega-3 fatty acids are possible adjunctive treatment options for patients with depression, as there is evidence they are more effective than placebo. There is not enough research to recommend omega-3 over antidepressants. Described mixed findings from the literature as to whether omega-3 is more effective as an adjunct or main therapy. Recommended dose was 1-2g daily (morning and evening doses). | Not reported |
| **Weatherby, Vital Choice, 2014** [63] | https://www.vitalchoice.com/article/omega-3s-linked-to-lower-teen-anxiety | Online article | United States | General public | Neutral | Research shows omega-3 may ameliorate anxiety in youth, although strong conclusions cannot be drawn at this point. Increased omega-3 intake is associated with increased levels of brain derived neurotrophic factor, which may improve brain plasticity and help regulate mood. | Not reported |

ADHD = Attention-Deficit with Hyperactivity Disorder; DHA = docosahexaenoic acid EPA = eicosapentaenoic acids.

*Gastrointestinal disturbances include heartburn, indigestion, nausea and diarrhoea.

evidence for its effectiveness. In terms of comprehensiveness, 33% of sources had an excellent amount of detail, including detail on contexts in which omega-3 was found to be effective, and potential mechanisms for its efficacy. However, 50% of sources provided a poor level of detail.

## Discussion

This scoping review was the first to summarise the existing evidence on the efficacy of omega-3 supplementation in treating depression and anxiety symptoms in young people aged 14–24. We found little evidence to support the use of omega-3 supplements in reducing depressive symptoms in young people, consistent with one previous meta-analysis which found that omega-3 supplements were not effective in treating depression among children aged 6–18 [9]. We also found some evidence that omega-3 supplements may reduce symptoms of anxiety, however, none of the included studies were conducted in populations with clinically diagnosed anxiety or depression. Studies were highly heterogenous in intervention characteristics (e.g., dosage) and participant characteristics (e.g., diagnosis) and there was no evidence to suggest that any such characteristics moderated the efficacy of omega-3 supplements. None of the included studies systematically examined mechanisms of action for omega-3, but two studies found evidence to suggest that omega-3 reduced inflammatory biomarkers which may be implicated in reduction of anxiety and depression symptoms, at least in populations without diagnosed mental illness [41, 46]. Our review also suggests that consumption of omega-3 supplements is not associated with significant side effects in young people. Compared with lack of evidence for supplement efficacy in peer-reviewed literature, grey literature information sources generally supported the use of omega-3 in reducing symptoms of depression and anxiety in young people. However, these sources differed in whether they recommended omega-3 be obtained through the diet or with the use of supplements, and they typically recommended lower dosages than were used in randomised controlled trials.

### Potential contextual moderators of omega-3 efficacy

There was little evidence to suggest that contextual variables such as baseline symptom severity, treatment characteristics (e.g., dosage, ratio of EPA to DHA, duration), whether omega-3 was administered alongside other treatments, or participant characteristics (e.g., sex, diagnosis) moderated the efficacy of omega-3 supplements. Moreover, no included studies were specifically designed to identify moderators of efficacy. However, research conducted among adults suggests that omega-3 may be more effective when administered as an adjunct to antidepressant medication, rather than as a standalone treatment [14, 21] and when EPA is administered at a higher dose [22].

With regard to treatment dosage and duration, the International Society for Nutritional Psychiatry Research Practice [24] recommends a daily dose of 1000-2000mg of omega-3 for at least 8 weeks. Based on a cut-off of 2000mg/day to distinguish between high and low doses [64], approximately half of the included RCTs in the current review administered a 'high' dose of omega-3 (> 2000mg/day [40, 44–47, 49, 50]) while the other half administered a 'low' dose (< 2000mg/day [39, 41–43, 48, 51]). We found no difference between high-dose and low-dose studies in terms of treatment efficacy. Furthermore, two studies that compared the efficacy of different doses did not find significant effects [36, 50]. There was also a wide range of treatment durations in the included studies. Notably, however, only two of the RCTs had a treatment duration of six months or longer [48, 50]. This is despite evidence showing that six months is the minimum period needed to ensure equilibration of omega-3 throughout the body [65]. As such, the absence of a clear effect of omega-3 supplementation on depression and anxiety symptoms may be due to an insufficient amount of time allowed for omega-3

supplements to reduce symptoms. Future studies should further investigate whether a certain dosage and duration of omega-3 supplementation is optimal to treat symptoms of depression and anxiety in young people.

There are a number of other factors not assessed in this review that may moderate efficacy of omega-3 supplementation in depression and anxiety. Previous research in adults with depression has found that omega-3 supplementation might be most effective for people with high red blood cell levels of EPA and DHA at baseline [66], however no included studies assessed this. Baseline ratios of omega-3 to omega-6 may also be relevant to risk of depression and anxiety due to their differing effects on inflammation [67, 68]. Specifically, omega-3 acids produce eicosanoids and related substances which suppress inflammation, while omega-6 acids produce eicosanoids and related substances that stimulate inflammation [69]. In this review, one study found that higher omega-3 to omega-6 ratios were associated with lower levels of anxiety- and depression-related symptoms [46]. As such, it may be of interest for future studies to examine how the efficacy of omega-3 supplementation in treating depression and/or anxiety symptoms may be moderated by baseline omega-3 levels and baseline ratio of omega-3 to omega-6.

As mentioned earlier, some previous meta-analyses have found that omega-3 supplements may reduce symptoms of both depression [8, 20–23] and anxiety [28] in adult populations, albeit the evidence is not of a sufficiently high quality [25]. Our findings raise the question of whether omega-3 supplementation may be even less effective for young people. Indeed, a recent longitudinal study found an association between baseline levels of omega-3 and omega-6 polyunsaturated fatty acids and symptoms of depression and anxiety in a 24-year-old cohort, but not for a 17-year-old cohort [70]. Age-dependent effects of omega-3 may relate to changes in the brain that occur during adolescence and young adulthood [71]. For instance, research in adult populations suggests that omega-3 supplements may be particularly effective as an adjunct to antidepressant treatment such as selective serotonin reuptake inhibitors (SSRIs) given that they also interact with serotonin receptors [14, 21]. However, adolescent brain development is characterised by lower expression of serotonin transporters, which may limit the potential adjunctive effect of omega-3 administered with SSRIs [72]. Longitudinal prospective studies that examine the effect of omega-3 over time are needed to better understand how age may moderate efficacy.

## Mismatch between peer-reviewed and grey literature

There was a clear discrepancy between the peer-reviewed and grey literature in the main message communicated about the efficacy of omega-3 in mitigating anxiety and depression among young people. Specifically, the grey literature overwhelmingly recommended the use of omega-3 for treating depression and anxiety, whereas the peer-reviewed literature reported scant evidence in support of this claim. Based on our ratings, grey literature sources that substantiated their claims with reference to peer-reviewed studies presented content that was more closely aligned with the scientific literature. To our knowledge, no other reviews on omega-3 supplements have assessed grey literature sources. However, our findings are consistent with a recent systematic review, which found that online health information intended for public consumption is generally poorly aligned with scientific evidence [73].

## Strengths and limitations

Strengths of the current scoping review included the broad scope, the use of a rigorous systematic search strategy, and the thorough assessment of the quality of eligible RCTs and grey literature sources. The assessment of whether grey literature sources aimed at the general public

accurately reflected the peer reviewed evidence is an important novel contribution of this review. The review was also enhanced by the involvement of a diverse group of stakeholders comprising of young people with lived experience of anxiety or depression, parents and care-givers, and healthcare professionals, which ensured that the review scope addressed research questions relevant to both lay and academic audiences.

The current review also had a several limitations. Only five studies in the review investigated anxiety outcomes, but none were in clinically anxious populations, and all but two RCTs had risk of bias from at least one source. The heterogeneity of studies included in the review made it difficult to determine whether participant characteristics (e.g., sex, diagnosis) or intervention characteristics (e.g., dosage and duration of treatment, adjunctive vs. standalone treatment) influenced the efficacy of omega-3 supplementation. Furthermore, although several RCTs in this review permitted continuation of current antidepressant medications and therapeutic treatments, they did not assess whether the type of treatment-as-usual that was combined with omega-3 was a moderator of the effect. Additionally, no studies systematically compared the effect of omega-3 supplements as a standalone treatment to their effect as an adjunct to other treatments. Finally, with the exception of one study, all studies included in this review were conducted in high income countries, and all stakeholders were Australian, which may limit the generalisability of findings to other populations.

## Implications for future research and policy

The findings of the current review have implications for clinical practice. Current practice guidelines relating to omega-3 supplements recommend a higher ratio of EPA to DHA and daily dosages that exceed 1000mg [24, 64]. However, these guidelines are not age-specific. Our review suggests that further research on the efficacy of omega-3 supplementation is warranted before it is recommended as a treatment for depression and anxiety symptoms in young people. In particular, it would be important for future studies to test whether omega-3 supplementation over a longer duration improves symptoms, given the duration of treatment in many studies included in this review may have been insufficient. Furthermore, trials should investigate different adjunct treatments or potential moderators to identify contexts under which omega-3 supplements are most effective to inform practitioners, and practice guidelines.

With the inclusion of grey literature, this scoping review also identified that online sources often did not appropriately represent the evidence of efficacy of omega-3 supplements for depression and anxiety in young people. Therefore, improving the accuracy and accessibility of evidence-based online health information about the effectiveness of current and emerging potential treatments for common mental illness should be a priority for policymakers. This is critical given that information from such sources can significantly impact help-seeking related beliefs and behaviours [73]. Concurrently, public health resources should be allocated to improve the health literacy of the general public including providing education to the general public on how they can ascertain the credibility of online health information.

## Conclusion

This scoping review of academic and grey literature is the first to synthesise the evidence on the efficacy of omega-3 supplementation in treating symptoms of depression and anxiety in young people. We found limited evidence that omega-3 supplementation reduces symptoms of depression, and some evidence supporting the efficacy of omega-3 in reducing symptoms of anxiety. No clear patterns emerged regarding whether the efficacy of omega-3 supplementation was moderated by such factors as dosage, ratio of EPA to DHA, participant characteristics and treatment duration. Additionally, the heterogeneity in sample demographics made

moderators difficult to identify. In contrast to peer-reviewed literature, most grey literature sources recommended omega-3 supplements to improve symptoms of anxiety and depression. Despite most grey literature sources including reference to peer-reviewed literature, few accurately described the evidence. Further research is needed to investigate specific mechanisms that might underlie omega-3 supplementation and to systematically test how factors such as dosage, duration of treatment, age, and clinical characteristics may moderate its effectiveness.

## Supporting information

**S1 Table. Preferred reporting items for Systematic reviews and Meta-Analyses extension for Scoping Reviews (PRISMA-ScR) checklist.**
(DOCX)

**S2 Table. Search terms for academic databases.**
(DOCX)

**S3 Table. Grey literature search terms and databases.**
(DOCX)

**S4 Table. Cochrane risk of bias ratings for randomised controlled trials (n = 13).**
(DOCX)

**S5 Table. Ratings of grey literature according to comprehensiveness, accuracy of information, and reference to peer-reviewed literature (n = 12).**
(DOCX)

**S6 Table. Characteristics of included non-randomised controlled trials (n = 4).**
(DOCX)

**S7 Table. Findings from non-randomised controlled trials investigating the effect of omega-3 supplementation on depression and/or anxiety (n = 4).**
(DOCX)

## Acknowledgments

We would like to acknowledge the key contributions made by our stakeholder advisory group in informing the scope and approach taken in this review. The valuable insights provided by this group of young people with lived experience, parents and carers, and health professionals shaped the search strategy and interpretation of the results of the review. We would also like to acknowledge Ms Helen Glover (Enlightened Consultants) for her expert facilitation of the stakeholder workshops.

## Author Contributions

**Conceptualization:** Natalie M. Reily, Samantha Tang.

**Data curation:** Natalie M. Reily, Samantha Tang, Daniel Z. Q. Gan, Veronica Sheanoda.

**Formal analysis:** Natalie M. Reily, Samantha Tang, Ashlee Negrone, Daniel Z. Q. Gan.

**Funding acquisition:** Natalie M. Reily, Samantha Tang, Helen Christensen.

**Methodology:** Natalie M. Reily, Samantha Tang, Ashlee Negrone, Veronica Sheanoda.

**Project administration:** Natalie M. Reily.

**Supervision:** Natalie M. Reily, Helen Christensen.

**Writing – original draft:** Natalie M. Reily, Samantha Tang, Ashlee Negrone, Daniel Z. Q. Gan.

**Writing – review & editing:** Natalie M. Reily, Samantha Tang, Ashlee Negrone, Daniel Z. Q. Gan, Veronica Sheanoda, Helen Christensen.

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
