## [Decision Letter · Decision Letter 0]

30 Jan 2023

PONE-D-22-32790

Omega-3 supplements in the prevention and treatment of youth depression and anxiety symptoms: A scoping review

PLOS ONE

Dear Dr. Reily,

Thank you for submitting your manuscript to PLOS ONE. After careful consideration, we feel that it has merit but does not fully meet PLOS ONE’s publication criteria as it currently stands. Therefore, we invite you to submit a revised version of the manuscript that addresses the points raised during the review process.

We look forward to receiving your revised manuscript.

Kind regards,

Anthony A. Olashore, MBCHB, FWACP

Academic Editor

PLOS ONE

Journal Requirements:

Additional Editor Comments:

Minor corrections are needed at this stage; however, you should give more information regarding the non-clinical sample. Based on some of the comments raised in the introduction and the method session, it would be good to revise the discussion. 

Reviewers' comments:

Reviewer's Responses to Questions

**Comments to the Author**

1. Is the manuscript technically sound, and do the data support the conclusions?

Reviewer #1: Yes

Reviewer #2: Partly

Reviewer #3: Yes

2. Has the statistical analysis been performed appropriately and rigorously? 

Reviewer #1: Yes

Reviewer #2: N/A

Reviewer #3: N/A

3. Have the authors made all data underlying the findings in their manuscript fully available?

Reviewer #1: Yes

Reviewer #2: Yes

Reviewer #3: Yes

4. Is the manuscript presented in an intelligible fashion and written in standard English?

Reviewer #1: Yes

Reviewer #2: Yes

Reviewer #3: Yes

5. Review Comments to the Author

Reviewer #1: The manuscript is technically sound, with methodology appropriate to the study objectives. The data presented are clearly in support of the conclusions.

The narrative review was comprehensive and detailed enough with clear criteria for the selection of the articles.

The reviewed studies were specified with all relevant data available for scrutiny.

The writing style and language was easy to read and understand.

Reviewer #2: General Comments

The subject-matter raised in this paper is quite relevant to the psychopharmacology of mood and anxiety disorders with particular reference to a segment of the patient population from which data regarding the outcomes on the use of omega-3 as treatment are scarce. It should be of interest to the international audience of the journal.

Title

The title is succinct and focuses on depression and anxiety “symptoms” which the review tries to address. However, this broad perspective of “symptoms” rather than “disorders” also particularly leads to a conflation of issues/findings which creates some confusion as to the research questions being investigated.

Abstract

The abstract does have reasonably sufficient information.

Introduction

The introduction appears generally well-written. The language is easy to follow. However, the authors stated in their opening statement as follows: “Depression and anxiety are among the most common mental illnesses in young people, with 50% of lifetime cases of mental illness beginning before age 14, and 75% beginning before age 24 (1).” This seems to suggest that the emphasis of the review is on the “disorders” rather than the “symptoms” while the entire review does the opposite. The import of this conflation is that the patient samples required to examine the research questions will be different in order to avoid undue confounding in results interpretation – depressive symptoms in different disorders may not necessarily meet the criteria for the clinical diagnosis of depressive disorder. The introduction has to focus more specifically on depressive symptoms in different disorders. Furthermore, the scope of the primary research question appears too broad for the pooling of data as you have in a systematic or scoping review. The authors state the primary aim as follows: “The primary aim of this review was therefore to synthesise the current literature on the efficacy of omega-3 supplements for depression and anxiety in young people, including potential mechanisms of action and moderators of efficacy." Very few studies were available (only 2) who reported on inflammatory changes as potential mechanisms of action and no specific studies were included that were focused on “moderators” of efficacy.

Methods

(i) “Sample populations varied across studies, and included non-clinical samples (n = 3), people with depression (n = 4), and people with, or at risk of, a mental illness other than depression or anxiety, including psychosis, schizophrenia, bipolar disorder, borderline personality disorder, and anorexia (n = 6).”

a. Further information is required regarding the “non-clinical samples”.

b. The authors should be clear on why they chose to include samples that were being treated for other mental health conditions – this increases the risk of confounding in the data interpretation. The authors should address this from the perspective or “symptomatic treatment” or treatment of “comorbidity” versus the treatment of a specific clinical disorder alone. I note that the authors indicated that this was a weakness in an earlier meta-analysis (Zhang L, Liu H, Kuang L, Meng H, Zhou X. Omega-3 fatty acids for the treatment of 527 depressive disorders in children and adolescents: A meta-analysis of randomized placebo528 controlled trials. Child Adolesc Psychiatry Ment Health. 2019;13:36). On the contrary, I think it is a strength in systematic reviews/meta-analysis to make the research question as specific as possible.

(ii) As stated above, very few studies were available (only 2) who reported on inflammatory changes as potential mechanisms of action and no specific studies were included that were focused on “moderators” of efficacy. Were there studies included in this review which focused specifically on identifying “moderators of efficacy”? If not, the current methodology would not be able to support any specific claims on moderators e.g. “Absence of evidence for contextual moderators of omega-3 efficacy”. Are there other studies out there which might have focused specifically on potential mechanisms of action of omega-3? This would either require a reworking of the research questions and the scope of the narrative synthesis or change the tone of the discussion of its limitations.

Discussion

The discussion should be revised as necessary based on comments raised about the introduction and methods.

References

References appear comprehensive and consistently presented. Some page numbers may be missing (or could be the due to the publication style of those materials).

Reviewer #3: An interesting paper which broadly investigated an area where there is dearth of literature.

The paper is well written and organized however, there are minor grammatical errors that the authors should take note of.

Introduction

- highlighted well the importance of undertaking the review.

- i found the argument that most young people do not benefit from pharmacotherapy and psychotherapy rather biased as there is evidence against it. I understand why the authors would want to present only one side. I believe it would be fair to present both sides and still justify the review.

Methods

- well articulated

- inclusion of the stakeholders is a welcome novel approach. May the authors kindly elaborate further on how they recruited the stakeholders, beyond just mentioning that they used Facebook

- Would it be possible to report Cohen's kappa as an indicator of coding reliability between the coders?

6. PLOS authors have the option to publish the peer review history of their article (what does this mean?). If published, this will include your full peer review and any attached files.

Reviewer #1: **Yes: **Olorunfemi Oladotun Ogunwobi

Reviewer #2: **Yes: **Adegboyega Ogunwale

Reviewer #3: No

---

## [Author Response · Author response to Decision Letter 0]

5 Feb 2023

Reviewer #1: The manuscript is technically sound, with methodology appropriate to the study objectives. The data presented are clearly in support of the conclusions.

The narrative review was comprehensive and detailed enough with clear criteria for the selection of the articles.

The reviewed studies were specified with all relevant data available for scrutiny.

The writing style and language was easy to read and understand.

Response: Thank you. 

 

Reviewer #2: General Comments

The subject-matter raised in this paper is quite relevant to the psychopharmacology of mood and anxiety disorders with particular reference to a segment of the patient population from which data regarding the outcomes on the use of omega-3 as treatment are scarce. It should be of interest to the international audience of the journal.

Response: Thank you. 

Title

The title is succinct and focuses on depression and anxiety “symptoms” which the review tries to address. However, this broad perspective of “symptoms” rather than “disorders” also particularly leads to a conflation of issues/findings which creates some confusion as to the research questions being investigated.

Response: Given the broad scope of the review, the word ‘symptoms’ was chosen to reflect that included studies used both clinical and non-clinical samples as there were no restrictions placed on psychiatric diagnoses. We have now modified the manuscript in order to ensure that the focus on symptoms is clear throughout. 

Abstract

The abstract does have reasonably sufficient information.

Response: Thank you.

Introduction

The introduction appears generally well-written. The language is easy to follow. However, the authors stated in their opening statement as follows: “Depression and anxiety are among the most common mental illnesses in young people, with 50% of lifetime cases of mental illness beginning before age 14, and 75% beginning before age 24 (1).” This seems to suggest that the emphasis of the review is on the “disorders” rather than the “symptoms” while the entire review does the opposite. The import of this conflation is that the patient samples required to examine the research questions will be different in order to avoid undue confounding in results interpretation – depressive symptoms in different disorders may not necessarily meet the criteria for the clinical diagnosis of depressive disorder. The introduction has to focus more specifically on depressive symptoms in different disorders. 

Response: Thank you for this comment. We have chosen to begin the manuscript with statistics about depressive and anxiety disorders despite our focus on symptoms given that symptoms of depression/anxiety can be indicative of or lead to a diagnosis of depression/anxiety. However, we have now clarified in our aims that we are examining the efficacy of omega-3 supplements for symptoms of depression and anxiety. We have also revised the introduction to highlight the high levels of comorbidity between depression and anxiety, and other mental health disorders. 

Furthermore, the scope of the primary research question appears too broad for the pooling of data as you have in a systematic or scoping review. 

Response: We chose to conduct a scoping review, rather than a systematic review given that scoping reviews are a preliminary assessment of the size and scope of a research topic, and are often broad in nature. In contrast, systematic reviews are characterised by their focus on a specific and narrow research question, using a systematic search and quality assessment to appraise and synthesise the quality of evidence available. Please see Grant and Booth (2009), A typology of Reviews (10.1111/j.1471-1842.2009.00848.x) for more information. 

The authors state the primary aim as follows: “The primary aim of this review was therefore to synthesise the current literature on the efficacy of omega-3 supplements for depression and anxiety in young people, including potential mechanisms of action and moderators of efficacy." Very few studies were available (only 2) who reported on inflammatory changes as potential mechanisms of action and no specific studies were included that were focused on “moderators” of efficacy.

Response: When designing review questions, it is standard practice to determine the aim(s) of the review a-priori, in order to shape the search strategy. Aims are not typically changed post-hoc based on the results of the search. The reviewer is correct in that only 2 studies report on potential mechanisms, and none report on moderators. Our review adds value by highlighting this gap in the existing literature. Throughout the discussion and in the conclusion of the manuscript we state that assessing potential moderators and mechanisms underlying omega-3 supplementation is an important avenue for future research. 

Methods

(i) “Sample populations varied across studies, and included non-clinical samples (n = 3), people with depression (n = 4), and people with, or at risk of, a mental illness other than depression or anxiety, including psychosis, schizophrenia, bipolar disorder, borderline personality disorder, and anorexia (n = 6).”

a0. Further information is required regarding the “non-clinical samples”.

Response: Further information regarding the sample demographic characteristics for the studies with non-clinical samples (and all other studies) has been provided in Table 1. Table 1 shows that the three non-clinical samples are: ‘healthy individuals’, ‘women with polycystic ovarian syndrome’ and ‘medical students’. We have now amended this sentence and included this additional information in the results section. 

b. The authors should be clear on why they chose to include samples that were being treated for other mental health conditions – this increases the risk of confounding in the data interpretation. 

The authors should address this from the perspective or “symptomatic treatment” or treatment of “comorbidity” versus the treatment of a specific clinical disorder alone. I note that the authors indicated that this was a weakness in an earlier meta-analysis (Zhang L, Liu H, Kuang L, Meng H, Zhou X. Omega-3 fatty acids for the treatment of 527 depressive disorders in children and adolescents: A meta-analysis of randomized placebo528 controlled trials. Child Adolesc Psychiatry Ment Health. 2019;13:36). On the contrary, I think it is a strength in systematic reviews/meta-analysis to make the research question as specific as possible.

Response: In the introduction section of the manuscript, we highlight that very little research has investigated whether omega-3 supplements can reduce symptoms of anxiety and depression in young people, despite some promising evidence of efficacy in adults. While the Zhang et al. systematic review and meta-analysis provides a high-quality assessment of the evidence for the efficacy of omega-3 supplements as a stand-alone depression treatment in children aged 6-18 with depression, it was not aiming to investigate the efficacy of omega-3 supplements more broadly in young people. As described in the introduction, the rationale for our review is to addresses this gap in the literature. We consider a broader age group, more diverse samples (including people without a mental illness and people with mental illnesses other than depression), and studies that assessed omega-3 as an adjunct rather than a primary, standalone treatment. While we agree that it is a strength for systematic reviews/meta-analyses to make the research question as specific as possible, we do not believe this to be the case for a scoping review (please see also earlier response on the difference between systematic and scoping reviews). Our scoping review is purposely broad in nature, and summarises the evidence for questions not covered by previous reviews. 

All included studies examine whether omega-3 supplementation had an effect on depression and/or anxiety symptoms. While including studies with diverse samples adds noise to the data, we disagree that it is a confounding variable given that we found no identifiable patterns based on participant diagnosis. Nonetheless, we have noted that the heterogeneity of included studies makes it difficult to determine whether participant characteristics such as diagnosis influenced the efficacy of omega-3 supplementation in the limitations section of the manuscript. We have also noted that further research is needed to more robustly determine whether diagnosis is a systematic moderator of omega-3 efficacy.

(ii) As stated above, very few studies were available (only 2) who reported on inflammatory changes as potential mechanisms of action and no specific studies were included that were focused on “moderators” of efficacy. Were there studies included in this review which focused specifically on identifying “moderators of efficacy”? If not, the current methodology would not be able to support any specific claims on moderators e.g. “Absence of evidence for contextual moderators of omega-3 efficacy”. 

Response: No included studies were focused specifically on identifying moderators of efficacy, which we have now noted in the revised version of the Results and Discussion. We had used the subheading ‘Absence of evidence for contextual moderators of omega-3 efficacy’ in the discussion to reflect that we found a dearth of published research examining potential moderators of the effect of omega-3 supplements on depression and/or anxiety in young people. We have now renamed this subheading to ‘Potential contextual moderators of omega-3 efficacy’ to improve the clarity of this header. 

Please note that we do not make any definitive claims about moderators of omega-3 efficacy in this section of the manuscript; rather, we have only speculated on potential moderators based on study characteristics and outcomes/variables assessed by included studies. 

Are there other studies out there which might have focused specifically on potential mechanisms of action of omega-3? This would either require a reworking of the research questions and the scope of the narrative synthesis or change the tone of the discussion of its limitations.

Response: There are several proposed mechanisms of action for omega-3 on symptoms of depression and anxiety, including inhibiting the inflammatory response via upregulation of proinflammatory cytokines and eicosanoids and upregulation of serotonin activity. We discuss these mechanisms in the second paragraph of the introduction (please see Berk et al., 2013; Simopoulos, 2002; Logan, 2003; Patrick & Ames, 2015, and Cowen, 2015, cited within the manuscript). For reviews focused on potential mechanisms see also Grosso et al., 2014 (depression), Polokowski et al., 2020 (anxiety). 

Studies that examined potential mechanisms of action for omega-3 in general were not in the scope of this review. We were specifically interested in understanding mechanisms of action underlying any effects of omega-3 supplements on depression and/or anxiety symptoms in young people. We believe any studies that examined both the effect of omega-3 supplements on symptoms of depression and/or anxiety in young people and mechanisms of action would have been detected by our search. Mechanisms specific to the effect on young people are of interest given that there may be age-dependent effects of omega-3 due to changes to the brain occurring during adolescence and young adulthood (see Mongan et al., 2021 and Johnson et al., 2009, referenced within the manuscript). We discuss this possibility within the fifth paragraph of the discussion. 

Discussion

The discussion should be revised as necessary based on comments raised about the introduction and methods.

Response: Please see above responses which describe changes made to the discussion section of the manuscript. 

References

References appear comprehensive and consistently presented. Some page numbers may be missing (or could be the due to the publication style of those materials).

Response: Articles without page numbers in the reference list are due to the publication style (online journal, no page numbers).

 

Reviewer #3: An interesting paper which broadly investigated an area where there is dearth of literature.

The paper is well written and organized however, there are minor grammatical errors that the authors should take note of.

Response: Thank you for this comment. We have now revised the manuscript to remove these grammatical errors. 

Introduction

- highlighted well the importance of undertaking the review.

Response: Thank you. 

- i found the argument that most young people do not benefit from pharmacotherapy and psychotherapy rather biased as there is evidence against it. I understand why the authors would want to present only one side. I believe it would be fair to present both sides and still justify the review.

Response: We did not state that ‘most’ young people do not benefit, rather that a ‘significant proportion’ do not benefit. Nonetheless we have now changed the language to ‘significant minority’ to reflect a more balanced view. 

Methods

- well articulated

- inclusion of the stakeholders is a welcome novel approach. May the authors kindly elaborate further on how they recruited the stakeholders, beyond just mentioning that they used Facebook

Response: Thank you. After posting the advertisement, potential stakeholders filled out an expression of interest that included questions about their basic demographic characteristics (age, gender), familiarity with omega-3 supplements, profession (for health professionals only) and availability to attend various workshop dates. We selected stakeholders on the basis of availability, as well to achieve a diverse group of stakeholders with regard to their demographic characteristics and experience/knowledge about omega-3 supplements. We have now some added additional detail on how stakeholders were recruited in the manuscript. 

- Would it be possible to report Cohen's kappa as an indicator of coding reliability between the coders?

Response: We now report Cohen’s kappa in the manuscript.

---

## [Decision Letter · Decision Letter 1]

8 Mar 2023

PONE-D-22-32790R1Omega-3 supplements in the prevention and treatment of youth depression and anxiety symptoms: A scoping reviewPLOS ONE

Dear Dr. Reily,

Thank you for submitting your manuscript to PLOS ONE. There are minor comments raised by one of the reviewers that I would like you to address before your manuscript can be considered. 

We look forward to receiving your revised manuscript.

Kind regards,

Anthony A. Olashore, PhD, FWACP

Academic Editor

PLOS ONE

Journal Requirements:

**Comments to the Author**

1. If the authors have adequately addressed your comments raised in a previous round of review and you feel that this manuscript is now acceptable for publication, you may indicate that here to bypass the “Comments to the Author” section, enter your conflict of interest statement in the “Confidential to Editor” section, and submit your "Accept" recommendation.

Reviewer #2: (No Response)

2. Is the manuscript technically sound, and do the data support the conclusions?

Reviewer #2: Yes

3. Has the statistical analysis been performed appropriately and rigorously? 

Reviewer #2: Yes

4. Have the authors made all data underlying the findings in their manuscript fully available?

Reviewer #2: Yes

5. Is the manuscript presented in an intelligible fashion and written in standard English?

Reviewer #2: Yes

6. Review Comments to the Author

Reviewer #2: My very warm thanks to the authors for their detailed responses to all my comments. I have only one minor amendment suggestion to the text of the paper:

p. 23, line 383-384: Since there were no included studies of moderators, it seems like an over-statement to imply that your findings contrast with those of studies among adults. Absence of studies is different from absence of evidence. Without the evidence, the claim of contrast cannot be upheld. I suggest that the statement be modified as: “However, research conducted among adults suggests that use of omega-3 as an adjunct to antidepressant medication and at higher doses may improve effectiveness thereby serving as moderators of treatment effect” [or something of that nature]. This avoids the burden of contrast while making the valid point.

7. PLOS authors have the option to publish the peer review history of their article (what does this mean?). If published, this will include your full peer review and any attached files.

Reviewer #2: **Yes: **Adegboyega Ogunwale

---

## [Author Response · Author response to Decision Letter 1]

15 Mar 2023

Reviewer #2: My very warm thanks to the authors for their detailed responses to all my comments. I have only one minor amendment suggestion to the text of the paper:

p. 23, line 383-384: Since there were no included studies of moderators, it seems like an over-statement to imply that your findings contrast with those of studies among adults. Absence of studies is different from absence of evidence. Without the evidence, the claim of contrast cannot be upheld. I suggest that the statement be modified as: “However, research conducted among adults suggests that use of omega-3 as an adjunct to antidepressant medication and at higher doses may improve effectiveness thereby serving as moderators of treatment effect” [or something of that nature]. This avoids the burden of contrast while making the valid point.

We thank Reviewer 2 for their feedback on our manuscript. We have now modified the statement appearing on p 23, in line with the author’s suggestion.

---

## [Editor Report · Decision Letter 2]

22 Mar 2023

Omega-3 supplements in the prevention and treatment of youth depression and anxiety symptoms: A scoping review

PONE-D-22-32790R2

Dear Dr. Reily,

We’re pleased to inform you that your manuscript has been judged scientifically suitable for publication and will be formally accepted for publication once it meets all outstanding technical requirements.

Kind regards,

Anthony A. Olashore, MBCHB, FWACP, PhD

Academic Editor

PLOS ONE

---

## [Editor Report · Acceptance letter]

3 Apr 2023

PONE-D-22-32790R2 

Omega-3 supplements in the prevention and treatment of youth depression and anxiety symptoms: A scoping review 

Dear Dr. Reily:

I'm pleased to inform you that your manuscript has been deemed suitable for publication in PLOS ONE. Congratulations! Your manuscript is now with our production department. 

Kind regards, 

on behalf of

Dr. Anthony A. Olashore 

Academic Editor

PLOS ONE